# Clinical Implications of Determining Individualized Positive End-Expiratory Pressure Using Electrical Impedance Tomography in Post-Cardiac Surgery Patients: A Prospective, Non-Randomized Interventional Study

**DOI:** 10.3390/jcm11113022

**Published:** 2022-05-27

**Authors:** Kiyoko Bito, Atsuko Shono, Shinya Kimura, Kazuto Maruta, Tadashi Omoto, Atsushi Aoki, Katsunori Oe, Toru Kotani

**Affiliations:** 1Department of Anesthesiology, School of Medicine, Showa University, 1-5-8 Hatanodai, Shinagawa-ku, Tokyo 142-8666, Japan; thunder.kimra@gmail.com (S.K.); oekanoekan@gmail.com (K.O.); 2Department of Intensive Care Medicine, School of Medicine, Showa University, 1-5-8 Hatanodai, Shinagawa-ku, Tokyo 142-8666, Japan; atsuko929shono@yahoo.co.jp (A.S.); trkotani@med.showa-u.ac.jp (T.K.); 3Department of Cardiovascular Surgery, School of Medicine, Showa University, 1-5-8 Hatanodai, Shinagawa-ku, Tokyo 142-8666, Japan; k-m227@med.showa-u.ac.jp (K.M.); tomoto@med.showa-u.ac.jp (T.O.); aokicvs@med.showa-u.ac.jp (A.A.)

**Keywords:** cardiac surgery, electrical impedance tomography, positive end-expiratory pressure

## Abstract

Optimal positive end-expiratory pressure (PEEP) can induce sustained lung function improvement. This prospective, non-randomized interventional study aimed to investigate the effect of individualized PEEP determined using electrical impedance tomography (EIT) in post-cardiac surgery patients (*n* = 35). Decremental PEEP trials were performed from 20 to 4 cmH_2_O in steps of 2 cmH_2_O, guided by EIT. PEEP levels preventing ventilation loss in dependent lung regions (PEEP_ONLINE_) were set. Ventilation distributions and oxygenation before the PEEP trial, and 5 min and 1 h after the PEEP_ONLINE_ setting were examined. Furthermore, we analyzed the saved impedance data offline to determine the PEEP levels that provided the best compromise between overdistended and collapsed lung (PEEP_ODCL_). Ventilation distributions of dependent regions increased at 5 min after the PEEP_ONLINE_ setting compared with those before the PEEP trial (mean ± standard deviation, 41.3 ± 8.5% vs. 49.1 ± 9.3%; *p* < 0.001), and were maintained at 1 h thereafter (48.7 ± 9.4%, *p* < 0.001). Oxygenation also showed sustained improvement. Rescue oxygen therapy (high-flow nasal cannula, noninvasive ventilation) after extubation was less frequent in patients with PEEP_ONLINE_ ≥ PEEP_ODCL_ than in those with PEEP_ONLINE_ < PEEP_ODCL_ (1/19 vs. 6/16; *p* = 0.018). EIT-guided individualized PEEP stabilized the improvement in ventilation distribution and oxygenation. Individual PEEP varies with EIT measures, and may differentially affect oxygenation after cardiac surgery.

## 1. Introduction

Mechanical ventilation can lead to lung damage in perioperative patients and the development of pulmonary complications [1,2]. Atelectasis, which commonly occurs after cardiac surgery [3], contributes to impaired oxygenation, and leads to tidal recruitment (atelectrauma) and overinflation of reduced normally-aerated lung regions (volutrauma/barotrauma) [4]. To reduce atelectasis and prevent ventilator-associated lung injury, the open lung approach, involving positive end-expiratory pressure (PEEP) subsequent to the recruitment maneuver (RM), is widely used [5,6,7,8]. Several bedside approaches, such as oxygenation, compliance, and pressure-volume curve, have been proposed to titrate the optimal PEEP. However, these parameters provide global information, without considering lung inhomogeneities.

Electrical impedance tomography (EIT) provides information on regional ventilation distribution by measuring breath-induced impedance changes [9,10]. EIT may help optimize PEEP settings and individualize mechanical ventilation [11,12]. Several EIT measures have been proposed to assess lung homogeneity [13,14,15,16], although the appropriate measures to optimize individualized PEEP settings and their clinical effects remain unclear. Tidal impedance variation (TIV) and end-expiratory lung impedance (EELI) values are provided by all commercially available EIT devices, and are used for bedside online analyses. Furthermore, the amount of overdistended and collapsed lung (ODCL) compartments identified via offline analysis using specialized software has been proposed as an individualized PEEP setting [17,18].

This study aimed to assess the clinical impact of EIT-guided individualized PEEP by prospectively examining the effects of individualized PEEP determined by online analysis (PEEP_ONLINE_) on the ventilation distributions, oxygenation, and respiratory mechanics in patients after cardiac surgery. Furthermore, we analyzed the saved impedance data offline to determine individualized PEEP levels using ODCL (PEEP_ODCL_), and assessed the effects of different methods of detecting individualized PEEP on ventilation distributions and oxygenation after extubation. Our results suggest that EIT-guided individualized PEEP settings can prevent deterioration in ventilation distribution and oxygenation over time, and that individualized PEEP varies with EIT measures and may differentially affect oxygenation after extubation.

## 2. Materials and Methods

### 2.1. Study Population

This prospective, non-randomized interventional study was conducted at Showa University Hospital in accordance with the Declaration of Helsinki and approved by the Medical Ethics Committee of Showa University School of Medicine (Showa University Ethical Committee, https://www.showa-u.ac.jp/research/ethics_committee/ethicsboard/index.html; protocol code 1686; date of approval, 22 January 2015). The study has been registered at UMIN Clinical Trial Registry under UMIN000017745. All patients provided written informed consent.

We included patients who had undergone elective cardiac surgery using cardiopulmonary bypass and were admitted to intensive care units between June 2015 and December 2017. The exclusion criteria were age <18 years, emergency surgery, hemodynamic instability (blood pressure <90 mmHg and/or heart rate <60 bpm), pneumothorax, and severe chronic obstructive pulmonary disease (Global Initiative for Obstructive Lung Disease stage III or IV).

### 2.2. Study Protocol

Intubated patients were mechanically ventilated with pressure-controlled ventilation (EVITA Infinity V500; Dräger Medical, Lübeck, Germany) at a PEEP of 6 cmH_2_O and an inspiratory oxygen fraction (FiO_2_) of 0.3–0.6. Driving pressure (plateau pressure—PEEP) was adjusted to deliver a tidal volume of 6–8 mL kg^−1^ of the predicted body weight. The respiratory rate was adjusted to achieve normocapnia while preventing auto-PEEP generation.

After hemodynamic stabilization, we increased PEEP to 20 cmH_2_O with a driving pressure of 12 cmH_2_O for 1 min. Thereafter, PEEP was decreased in steps of 2 cmH_2_O every 5 min until it reached 4 cmH_2_O. After increasing PEEP again to 20 cmH_2_O, it was set at a specific level, as described in Section 2.3. EIT results and respiratory compliance (tidal volume divided by driving pressure) were assessed at each step (before the PEEP trial, at the end of each PEEP step, and 5 min and 1 h after the PEEP setting). Thereafter, the driving pressure was set to deliver the above-mentioned target tidal volume. The PEEP level was maintained until ventilator weaning. After extubation, patients received standard oxygen therapy via face masks. Rescue oxygen therapy (high-flow nasal cannula oxygen therapy and noninvasive ventilation) was applied if the percutaneous arterial oxygen saturation was ≤90% with supplemental oxygen of ≥6 L min^−1^ and/or the respiratory rate was >30 breaths min^−1^ with the use of accessory muscles for respiration. The rescue oxygen therapies required were recorded. Arterial blood gas levels were determined before the PEEP trial, 5 min and 1 h after the PEEP setting, and after extubation, as required.

### 2.3. Individualized PEEP Setting by EIT (PEEP_ONLINE_)

An EIT electrode belt containing 16 electrodes was placed around the patient’s thorax at the fifth or sixth intercostal space level and connected to an EIT monitor (PulmoVista^®^ 500, software version 1.n; Dräger Medical, Lübeck, Germany). EIT lung images comprising a matrix of 32 × 32 pixels were divided into four equal regions of interest (ROIs) in the ventral-to-dorsal orientation, with ROIs 1 and 2 indicating the nondependent regions, and ROIs 3 and 4 indicating the dependent regions. Tidal images represented the regional distribution of ventilation-induced impedance changes. Differential images represented the difference between tidal images of the acquired and reference data. Regional differences were displayed as color-coded images, as shown in Figure 1. The global impedance waveform showed relative impedance changes in the entire electrode plane, and was displayed as a compressed waveform for the observation of EELI trends, indicating end-expiratory lung volume (EELV) changes.

A 20-cmH_2_O PEEP section was defined as the reference, and we assessed whether ventilation loss was evident in the dependent region on differential images in each PEEP section when compared with the reference. PEEP_ONLINE_ was defined as one step above the PEEP level at which ventilation loss occurred in the dependent region. If such a loss did not occur at any PEEP level, PEEP_ONLINE_ was defined as one step above the PEEP level with a downward slope of EELI. After increasing PEEP to 20 cmH_2_O, PEEP was set at PEEP_ONLINE_. If ventilation loss was evident in the dependent region on differential images in the PEEP section when compared with the second PEEP 20-cmH_2_O section, PEEP_ONLINE_ was increased by 1 cmH_2_O. Once PEEP_ONLINE_ was set, it was maintained until ventilator weaning. The individual level with maximal compliance during the decremental PEEP trial was defined as PEEP_COMPLIANCE_.

### 2.4. Offline EIT Data Analysis

EIT data were recorded for 2 min at the end of each PEEP step for subsequent offline analysis using EITdiag (Dräger Medical). Global TIV was calculated from impedance changes between end-inspiration and end-expiration in all lung regions, and the ventilation distribution in the dependent region was expressed as a percentage of the global TIV.

ODCL values (%) were calculated using the method described by Costa et al. [19]. Briefly, regional compliance at each EIT pixel was computed during a decremental PEEP trial, and its best compliance across all PEEP levels was estimated. We compared current pixel compliance with the best pixel compliance to calculate the percentages of overdistention and collapse per pixel at each PEEP level. Another individualized PEEP_ODCL_ was defined as the nearest PEEP above the crossing of cumulated overdistention and collapse percentage curves, providing the best compromise between ODCL areas.

EELI changes (ΔEELI) referring to TIV before the PEEP trial were also calculated. A change of 100% represents an EELI change of the same magnitude at the TIV before the PEEP trial. EELV changes (ΔEELV) occurring within 1 h after the PEEP setting were calculated based on the change in EELI from 5 min to 1 h after the PEEP setting. ΔEELV values were obtained by multiplying the change in ΔEELI from 5 min to 1 h after the PEEP setting with the ratio between the tidal volume and corresponding TIV (both measured 5 min after PEEP setting) [20].

The primary outcome of our study was the effect of PEEP_ONLINE_ on ventilation distribution in the dependent region. Secondary outcomes were the ratio of arterial oxygen partial pressure (PaO_2_) to FiO_2_ (PaO_2_/FiO_2_), and respiratory compliance. Additionally, to examine whether differences in individualized PEEP with EIT measures affected the same parameters and oxygen therapy after extubation, we conducted a post-hoc analysis to compare the data between patients with PEEP_ONLINE_ ≥ PEEP_ODCL_ and those with PEEP_ONLINE_ < PEEP_ODCL_.

### 2.5. Statistical Analyses

Based on the findings of Karsten et al. [21], the data of at least 25 patients are needed to detect a 3% change in TIV in the dependent region, with an alpha level of 0.05 and a standard deviation of 10%, using a repeated measures analysis of variance (ANOVA) at a power of 90%. Accordingly, we initially enrolled 40 patients, considering an estimated dropout of approximately one-third of the patients before the end of the study.

Continuous variables were tested for a normal distribution using the Shapiro–Wilk test. Normally distributed data are summarized as mean ± standard deviation, whereas skewed variables are summarized as median (interquartile range). Categorical data are expressed as numbers. Changes in the PaO_2_/FIO_2_ ratio, EIT data, and hemodynamic variables among the three time points (before the PEEP trial, and 5 min and 1 h after PEEP setting) were compared using a repeated measures ANOVA followed by post-hoc Bonferroni-corrected paired *t*-tests. A Friedman ANOVA for repeated measures was used to compare changes in respiratory compliance among time points, and pairwise comparisons using the Wilcoxon test with a Bonferroni correction were performed. Means were compared using unpaired *t*-tests, and medians were compared using the Wilcoxon test between patients with PEEP_ONLINE_ ≥ PEEP_ODCL_ and those with PEEP_ONLINE_ < PEEP_ODCL_. Categorical data were compared using the Chi-squared or Fisher’s exact test. All tests were two-tailed, and *p*-values <0.05 were considered significant. Statistical analyses were performed using SPSS (version 27; IBM Corp., Armonk, NY, USA).

## 3. Results

### 3.1. Patient Characteristics

Forty patients were initially enrolled in this study. However, PEEP decreased in five patients because of hemodynamic instability; thus, their data were excluded. Consequently, we included 35 patients; their characteristics are listed in Table 1.

### 3.2. Ventilation Distribution during the PEEP Trial

In 26 patients, ventilation was mainly distributed to the dependent region at 20 cmH_2_O PEEP, and ventilation to the nondependent region increased in response to lowering PEEP. Both regions were evenly ventilated at a PEEP of 10 (8–12) cmH_2_O. At all PEEP levels, ventilation was mainly distributed to the nondependent and dependent regions in six and three patients, respectively.

### 3.3. Effects of PEEP_ONLINE_

The PEEP_ONLINE_ was 10 (8–11) cmH_2_O. In 32 patients, we detected PEEP_ONLINE_ by ventilation loss in the dependent region during decremental PEEP trials. In the remaining three patients without ventilation loss in the dependent region, we determined PEEP_ONLINE_ based on a downward EELI slope. PEEP_COMPLIANCE_ was 6 (6–8) cmH_2_O.

Compared with the TIV before the trial, the TIV in the dependent region increased significantly 5 min after PEEP_ONLINE_ setting; however, no significant decrease occurred 1 h after PEEP_ONLINE_ setting (before PEEP_ONLINE_ setting: 41.3 ± 8.5%; 5 min after PEEP_ONLINE_ setting: 49.1 ± 9.3%; difference, 7.5%; 95% confidence interval (CI), 5.0 to 9.9%; *p* < 0.001; and 1 h after PEEP_ONLINE_ setting: 48.7 ± 9.4%; difference, 6.9%; 95% CI, 4.4 to 9.4%; *p* < 0.001). The PaO_2_/FiO_2_ ratio also showed sustained improvement (before: 310.3 ± 93.4; 5 min after: 428.7 ± 119.1 mmHg; difference, 118.4 mmHg; 95% CI, 90.9 to 146.0 mmHg; *p* < 0.001; and 1 h after: 429.9 ± 115.9 mmHg; difference, 119.6 mmHg; 95% CI, 92.1 to 147.2 mmHg; *p* < 0.001). Respiratory compliance increased 5 min after PEEP_ONLINE_ setting, followed by a decrease 1 h after; however, it remained higher than that before the PEEP trial. ΔEELI increased 5 min after PEEP_ONLINE_ setting and was maintained 1 h after PEEP_ONLINE_ setting (Table 2). The ΔEELV after PEEP setting (EELV change from 5 min to 1 h after the trial) was −70.9 ± 103.4 mL. Patients were extubated at 16.3 (8.7–19.2) h after intensive care unit admission. Although none required reintubation, seven (20%) patients received rescue oxygen therapy after extubation.

### 3.4. Comparison between PEEP_ONLINE_ and PEEP_ODCL_

We could detect the individual PEEP_ODCL_ in all patients. The PEEP_ODCL_ was 10 (8–12) cmH_2_O. The PEEP_ONLINE_ did not significantly differ from the PEEP_ODCL_, although these values were applicable in only seven (20%) patients. In 19 and 16 patients, PEEP_ONLINE_ was higher and lower, respectively, than PEEP_ODCL_. Three patterns in which PEEP_ONLINE_ and PEEP_ODCL_ did not correspond are shown in Figure 2. Both PEEP_ODCL_ and PEEP_ONLINE_ were significantly higher than PEEP_COMPLIANCE_ (*p* = 0.001 and *p* < 0.001, respectively).

ΔEELV values were lower in patients with PEEP_ONLINE_ ≥ PEEP_ODCL_ than in those with PEEP_ONLINE_ < PEEP_ODCL_ (−27.2 ± 87.7 vs. −122.8 ± 98.6; difference, 95.6; 95% CI, 31.6 to 159.7; *p* = 0.005). Moreover, patients with PEEP_ONLINE_ ≥ PEEP_ODCL_ required less rescue oxygen therapy after extubation than patients with PEEP_ONLINE_ < PEEP_ODCL_ (1/19 and 6/16, respectively; odds ratio, 0.09; 95% CI, 0.01 to 0.882; *p* = 0.018) (Table 3). Patient characteristics were comparable between PEEP_ONLINE_ ≥ PEEP_ODCL_ and PEEP_ONLINE_ < PEEP_ODCL_ groups (Table 4).

## 4. Discussion

In the present study, the PEEP_ONLINE_ setting (i.e., PEEP setting in the EIT online analysis that avoided ventilation loss in the dependent lung region when compared with 20 cmH_2_O) prevented lung collapse progression and ensured improved oxygenation and respiratory compliance. In 80% of patients, PEEP_ONLINE_ did not correspond with the EIT-guided offline-determined PEEP, which provided the best compromise between overdistention and collapse (PEEP_ODCL_). The decrease in EELV was lower in patients with PEEP_ONLINE_ ≥ PEEP_ODCL_ than in those with PEEP_ONLINE_ < PEEP_ODCL_, and the former required less rescue oxygen therapy after extubation.

An RM increases the lung area available for ventilation, and consequently improves oxygenation and lung mechanics. However, the appropriate method of setting the optimal PEEP individually to preserve the beneficial effects after the RM remains controversial. Sufficiently high levels of PEEP are required to prevent derecruitment, although this incurs the potential risk of overdistention. In the present study, the PEEP_ONLINE_ was set at the minimum required pressure to prevent derecruitment in the dependent region compared with 20 cmH_2_O, thereby reducing the risk of overdistention in the nondependent region. Patients at high risk of postoperative pulmonary complications, such as cardiovascular, thoracic, and abdominal surgery, may benefit from EIT-guided individualized PEEP, which help ensure lung protective ventilation and improve pulmonary function.

The PEEP_ONLINE_ was 10 (8–11) cmH_2_O, which corresponded approximately with the “best” PEEP level after cardiac surgery determined by EIT in previous studies [11,21]. Furthermore, the dependent and nondependent regions were almost evenly ventilated in PEEP_ONLINE_, and this homogeneous ventilation was maintained for 1 h, along with the oxygenation. In a study using an acute respiratory distress syndrome (ARDS) animal model, EIT-guided mechanical ventilation, used to maximize the recruitment of the dependent lung and minimize the overdistention of nondependent lung areas, improved oxygenation and respiratory mechanics, and decreased ventilator-associated lung injury [12].

Postoperative pulmonary complications occur in approximately 7–33% of patients after cardiac surgery [22,23]. In the present study, seven (20%) patients required rescue oxygen therapy after extubation, although none required reintubation. Moreover, the ΔEELV at 1 h after PEEP setting was smaller, and rescue oxygen therapy was required less frequently in patients with PEEP_ONLINE_ ≥ PEEP_ODCL_ than in patients with PEEP_ONLINE_ < PEEP_ODCL_.

The effectiveness of EIT-guided PEEP using ODCL has been previously discussed. In patients with severe ARDS, PEEP titration according to ODCL may improve oxygenation, compliance, and the success rate of weaning from the ventilator more effectively than the pressure-volume curve [17]. In patients under general anesthesia, the application of PEEP obtained with ODCL not only improved the driving pressure and oxygenation, but also reduced postoperative atelectasis [18]. Similarly, our findings suggested that applying PEEP ≥ PEEP_ODCL_ prevented derecruitment, and preserved lung function until after extubation.

For various reasons, PEEP_ODCL_ did not always correspond with PEEP_ONLINE_. First, when ventilation in the dependent region at a reference PEEP of 20 cmH_2_O decreased, owing to high overdistention with low tidal impedance change, the capability to detect ventilation loss due to lung collapse with lowering PEEP might have been less sensitive (Figure 2a). Second, when the decreases in the PEEP simultaneously caused focal ventilation losses and gains within the same dependent region, PEEP_ONLINE_ was higher than PEEP_ODCL_ (Figure 2b). Third, when ventilation in the dependent region was almost unchanged in the PEEP trials, ventilation loss was difficult to detect, impeding PEEP_ONLINE_ determination (Figure 2c). In such cases, online analysis was unsuitable for setting individualized PEEP, whereas offline analysis using ODCL was highly sensitive to ventilation loss because the best pixel compliance across all PEEP levels was used as a reference. Moreover, individualized PEEP_ODCL_ was obtained in all patients, indicating its feasibility.

Both PEEP_ONLINE_ and PEEP_ODCL_ were higher than PEEP_COMPLIANCE_. Individualized EIT-guided PEEP does not always correspond with the PEEP providing the best compliance [21,24]. One reason is that compliance is derived from global lung parameters, whereas EIT measures reflect regional information. During decremental PEEP trials, ventilation loss caused by the collapse of dependent regions, and ventilation gain caused by the relief of overdistention in nondependent regions occur simultaneously. If the ventilation increase due to the relief of overdistention is higher than the collapse-induced ventilation decrease, global compliance increases [25]. Moreover, the emergence of tidal recruitment cannot be detected when assessing compliance changes alone [26]. Whole-lung computed tomography scans of patients with ARDS who underwent decremental PEEP titration revealed that the collapse in dependent regions began at approximately 4 cmH_2_O above the PEEP of maximal compliance [27]. Further, the EIT measurement includes only one cross-sectional slice of the thorax, and cannot detect the whole ventilation distribution in the craniocaudal direction. We measured EIT at the level just above the diaphragm where maximum atelectasis is expected to occur, although overdistention may be underestimated compared with that assessed using measurements at the cranial level [28].

Our study has some limitations. To avoid serious hemodynamic impairment directly after cardiac surgery, PEEP was limited to 20 cmH_2_O, which was lower than that in RMs usually performed in patients with ARDS. The dependent region was not predominantly ventilated during PEEP trials in six patients, indicating that the dependent region was not fully recruited. However, in these patients, ventilation in the dependent regions remained almost unchanged regardless of the PEEP level (Figure 2c), whereas further recruitment with airway pressures considered clinically acceptable was not expected. Second, we monitored the effect of PEEP_ONLINE_ for only 1 h. In previous studies assessing changes in PEEP effects over time, the gain in oxygenation obtained using RM was lost within 15–30 min when the RM was not followed by a sufficiently high PEEP [29,30]. Therefore, 1 h of observation after PEEP settings was considered sufficient to assess whether derecruitment occurred. The offline analysis was not available at the bedside; thus, we could not conduct a crossover or prospective randomized trial_._ Currently, online analysis of ODCL can be performed using the latest version of the EIT monitor (software version 1.2n); further randomized controlled trials are expected. Finally, we recruited a relatively limited number of patients; therefore, a large-scale clinical study is required to assess whether EIT-determined individualized PEEP values are beneficial to determine patient outcomes.

Overall, after cardiac surgery, EIT-guided individualized PEEP settings may prevent deterioration in ventilation distribution and oxygenation over time. Individualized PEEP varies with EIT measures, and may differentially affect oxygenation after extubation.

## Figures and Tables

**Figure 1 jcm-11-03022-f001:**
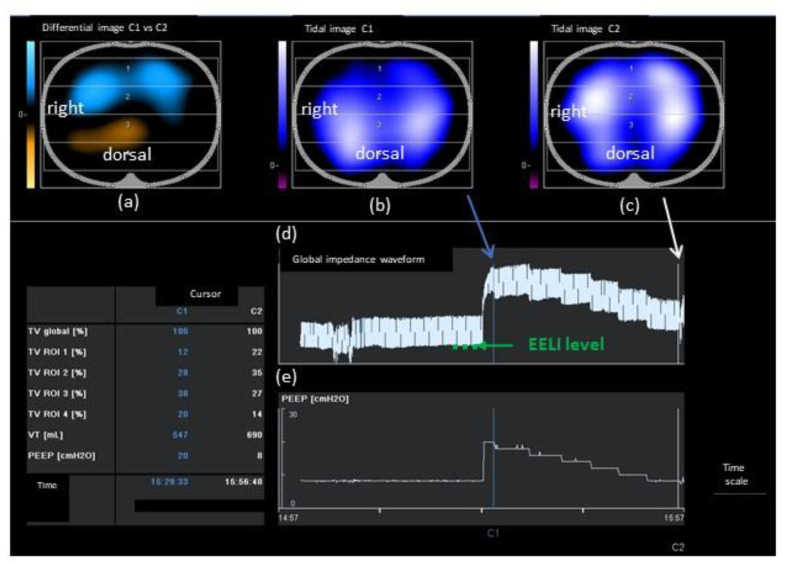
An example of EIT online analysis. (**a**) Differential image between PEEP 8 cmH_2_O and PEEP 20 cmH_2_O, defined as the reference. Regional differences are displayed as color-coded images, with blue indicating increases, and orange indicating decreases, compared with reference data. (**b**) Tidal image at PEEP 20 cmH_2_O. Regional ventilations are displayed as color-coded images. Bright-colored regions (corresponding to large impedance) represent well-ventilated areas. Dark-colored regions (small impedance change) represent less-ventilated areas. (**c**) Tidal image at PEEP 8 cmH_2_O. (**d**) Global impedance waveform. The dotted line indicates the EELI level. (**e**) PEEP level. EIT, electrical impedance tomography; PEEP, positive end-expiratory pressure; EELI, end-expiratory lung impedance; ROI, region of interest; TV, tidal impedance variation; VT, tidal volume.

**Figure 2 jcm-11-03022-f002:**
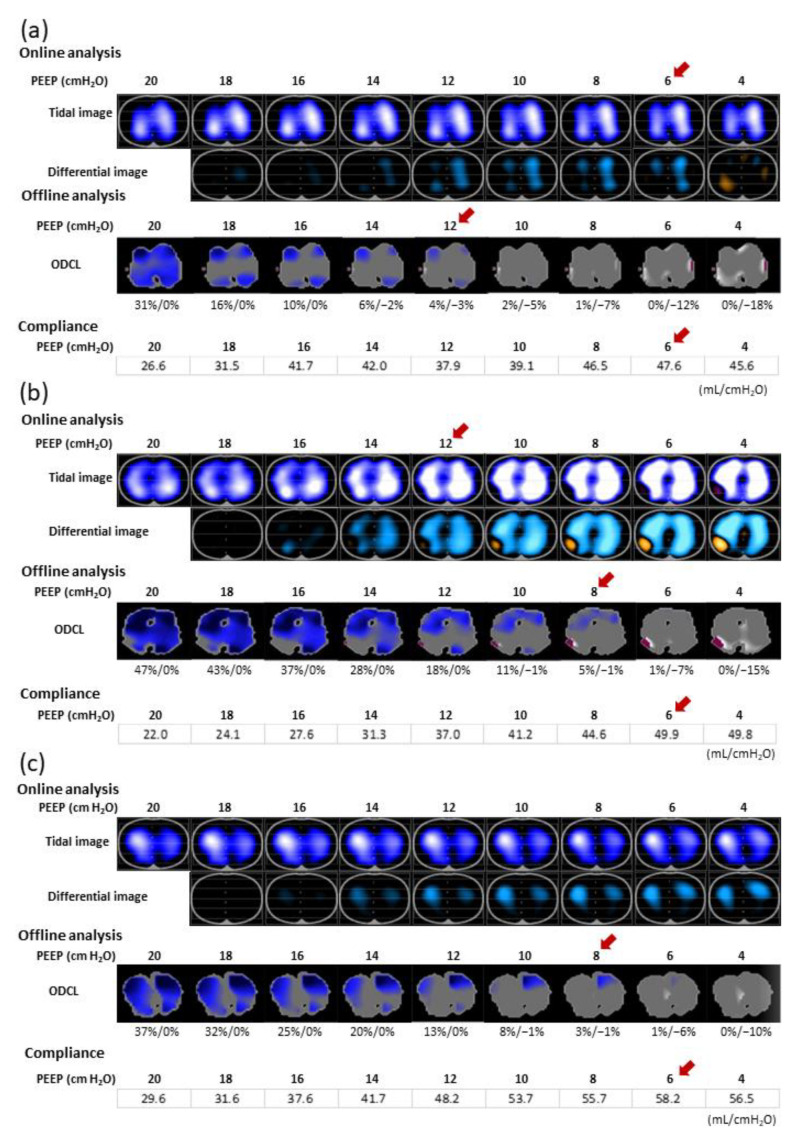
Ventilation distribution pattern 1 in which PEEP_ODCL_ does not correspond with PEEP_ONLINE_. (**a**) In the tidal images of the online analysis, bright-colored regions represent well-ventilated areas, whereas dark-colored areas represent less-ventilated areas. In the differential images of the online analysis, regional differences displayed in blue, and orange indicate increases and decreases, respectively, compared with reference data. Ventilation loss (orange) is evident in the dependent region at PEEP 4 cmH_2_O in the differential image. PEEP_ONLINE_ is 6 cmH_2_O. In ODCL images of the offline EIT analysis, blue-colored regions indicate overdistention, and white-colored regions indicate collapse. PEEP_ODCL_ is 14 cmH_2_O, which differs substantially from PEEP_ONLINE_. PEEP_COMPLIANCE_ is 6 cmH_2_O. (**b**) In differential images of the online analysis, focal ventilation loss on the right side in the dependent lung region and ventilation increase occur simultaneously in the same dependent lung region when lowering PEEP. PEEP_ONLINE_ is 12 cmH_2_O. In the offline analysis, PEEP_ODCL_ is 8 cmH_2_O, which differs substantially from PEEP_ONLINE_. PEEP_COMPLIANCE_ is 6 cmH_2_O. (**c**) In tidal images of the online analysis, ventilation in the dependent lung region remains low and almost unchanged, regardless of the PEEP level. Ventilation loss cannot be detected at any PEEP level in differential images; thus, PEEP_ONLINE_ cannot be determined based on ventilation loss in the dependent region. In the offline analysis, PEEP_ODCL_ is calculated as 8 cmH_2_O. PEEP_COMPLIANCE_ is 6 cmH_2_O. EIT, electrical impedance tomography; ODCL, overdistended and collapsed lung; PEEP, positive end-expiratory pressure; PEEP_ODCL_, PEEP levels using ODCL by offline analysis; PEEP_ONLINE_, PEEP determined by online analysis; PEEP_COMPLIANCE_, PEEP with maximal compliance during the PEEP trial.

**Table 1 jcm-11-03022-t001:** Patient characteristics.

Characteristic	Value
Number of patients	35
Age (years)	71.3 ± 10.0
Sex (female/male)	18/17
Height (cm)	155.2 ± 10.0
Weight (kg)	58.3 ± 13.2
Body mass index (kg m^−2^)	24.1 ± 4.5
Duration of CPB (min)	95 (65–127)
Duration of surgery (min)	224.3 ± 47.5
Comorbidity	
Hypertension	22
Diabetes	7
Chronic kidney disease	10
EuroSCORE II	2.0 (1.4–3.5)
Type of surgery	
Aortic valve	17
Mitral valve	2
Combination	13
Others	3

Data are presented as mean ± standard deviation, median (interquartile range), and numbers. CPB, cardiopulmonary bypass; EuroSCORE II, European System for Cardiac Operative Risk Evaluation II score.

**Table 2 jcm-11-03022-t002:** Changes in respiratory and EIT parameters.

Parameter	Before PEEP Trial	5 min after PEEP_ONLINE_ Setting	1 h after PEEP_ONLINE_ Setting	*p*-Value
TIV in the dependent region (%)	41.3 ± 8.5	49.1 ± 9.3 *	48.7 ± 9.4 *	<0.001
ΔEELI (%)	0.0	142.7 ± 109.4 *	125.5 ± 119.9 *	<0.001
PaO_2_/FiO_2_ ratio	310.3 ± 93.4	428.7 ± 119.1 *	429.9 ± 115.9 *	<0.001
Respiratory compliance (mL cmH_2_O^−1^)	41.3 (32.8–53.0)	46.8 (39.8–63.8) *	47.0 (35.8–61.1) *^,^^☨^	<0.001

Data are presented as mean ± standard deviation and median (interquartile range). * A significant difference compared with before the PEEP trial (*p* < 0.01). ^☨^ A significant difference compared with 5 min after PEEP_ONLINE_ setting. ΔEELI, changes in end-expiratory lung impedance; EIT, electrical impedance tomography; PaO_2_/FiO_2_ ratio, the ratio of arterial oxygen partial pressure to inspiratory oxygen function; PEEP, positive end-expiratory pressure; PEEP_ONLINE_, PEEP determined by online analysis; TIV, tidal impedance variation.

**Table 3 jcm-11-03022-t003:** Comparison between PEEP_ONLINE_ ≥ PEEP_ODCL_ and PEEP_ONLINE_ < PEEP_ODCL_ groups.

Parameter	PEEP_ONLINE_ ≥ PEEP_ODCL_ (*n* = 19)	PEEP_ONLINE_ < PEEP_ODCL_ (*n* = 16)	*p*-Value
PEEP_ONLINE_ (cmH_2_O)	10 (9–12)	7.5 (6–10)	0.008
PEEP_ODCL_ (cmH_2_O)	10 (8–10)	10.5 (8.5–12) ^#^	0.047
PEEP_COMPLIANCE_ (cmH_2_O)	6 (6–8) ^#^	8 (6–9.5)	0.106
TIV in the dependent region (%)			
before PEEP trial	41.8 ± 9.1	40.8 ± 7.8	0.753
5 min after PEEP_ONLINE_ setting	50.9 ± 9.9 *	47 ± 8.4 *	0.229
1 h after PEEP_ONLINE_ setting	50.6 ± 10.1 *	46.4 ± 8.3 *	0.189
PaO_2_/FiO_2_ ratio			
before PEEP trial	322.1 ± 86.1	296.3 ± 102.5	0.424
5 min after PEEP_ONLINE_ setting	456.4 ± 116.5 *	395.8 ± 117.2 *	0.136
1 h after PEEP_ONLINE_ setting	456.5 ± 111.4 *	398.5 ± 110.4 *	0.143
Respiratory compliance (mL cmH_2_O^−1^)			
before PEEP trial	41.3 (35.9–55.4)	42.5 (31.4–51.1)	0.417
5 min after PEEP_ONLINE_ setting	52.6 (40.6–78.1) *	44.7 (39–59.4) *	0.267
1 h after PEEP_ONLINE_ setting	48.1 (37.3–71.3) *	42.5 (34.3–58.9)	0.202
ΔEELV after PEEP_ONLINE_ setting (mL)	−27.2 ± 87.7	−122.8 ± 98.6	0.005
Lowest PaO_2_/FiO_2_ ratio within 12 h after extubation (mmHg)	214.5 ± 62.7	165.0 ± 76.1	0.034
Patients who required rescue oxygen therapy (*n*)	1	6	0.018
Patients with pneumonia (*n*)	1	1	0.900

Data are presented as mean ± standard deviation, median (interquartile range), and numbers. ^#^ A significant difference compared with PEEP_ONLINE_ (*p* < 0.01). * A significant difference compared with before the PEEP trial (*p* < 0.01). ΔEELV, changes in end-expiratory lung volume; PaO_2_/FiO_2_ ratio, the ratio of arterial oxygen partial pressure to inspiratory oxygen function; PEEP, positive end-expiratory pressure; PEEP_COMPLIANCE_, PEEP with maximal compliance during the PEEP trial; PEEP_ODCL_, PEEP levels using overdistended and collapsed lung by offline analysis; PEEP_ONLINE_, PEEP determined by online analysis; TIV, tidal impedance variation.

**Table 4 jcm-11-03022-t004:** Characteristics of patients with PEEP_ONLINE_ ≥ PEEP_ODCL_ and PEEP_ONLINE_ < PEEP_ODCL_.

Characteristic	PEEP_ONLINE_ ≥ PEEP_ODCL_ (*n* = 19)	PEEP_ONLINE_ < PEEP_ODCL_ (*n* = 16)	*p*-Value
Age (years)	71 ± 8.7	71.6 ± 11.7	0.872
Sex (female/male)	9/10	8/8	0.877
Height (cm)	155.9 ± 9.0	154.3 ± 11.4	0.632
Weight (kg)	58.5 ± 10.4	58.0 ± 16.3	0.908
BMI (kg m^−2^)	24.1 ± 4.2	24.1 ± 5.0	0.989
EuroSCORE II	1.9 (1.0–3.8)	2.4 (1.7–3.5)	0.275
Type of surgery			0.302
Aortic valve	11	6	
Mitral valve	0	2	
Combination	6	7	
Others	2	1	
Duration of CPB (min)	92.5 ± 43.7	110.6 ± 31.5	0.178
Duration of surgery (min)	211.6 ± 43.6	239.3 ± 48.8	0.085
Time of extubation after ICU admission (h)	16.3 (8.7–19.2)	16.3 (9.1–19.7)	0.778
Patients who underwent blood transfusion (*n*)	10	11	0.332

Data are presented as mean ± standard deviation, median (interquartile range), and numbers. PEEP, positive end-expiratory pressure; PEEP_ONLINE_, PEEP determined by online analysis; ODCL, overdistended and collapsed lung; BMI, body mass index; EuroSCORE II, European System for Cardiac Operative Risk Evaluation II; CPB, cardiopulmonary bypass; ICU, intensive care unit.

## Data Availability

The datasets used and/or analyzed during the current study are available from the corresponding author upon reasonable request. The data are not publicly available due to their containing information that could compromise the privacy of research participants.

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
