# Peer review of "Clinical Implications of Determining Individualized Positive End-Expiratory Pressure Using Electrical Impedance Tomography in Post-Cardiac Surgery Patients: A Prospective, Non-Randomized Interventional Study"

_jcm, 2022, doi:10.3390/jcm11113022_

Round 1
Reviewer 1 Report
The manuscript under review is an interesting interventional study aimed to assess the role of individualized PEEP levels, evaluated by EIT, in lung function improvement in patients who underwent cardiac surgery. The study is well conducted, the subject selection and methods are well explain. The discussion is extensive and detailed, the conclusions are consistent with the results. I only have a few minor comments.
- In table 1, it would be useful to add the patients comorbidities in the “characteristic” column, if these data are available.
- The reference list should be implemented with the following papers: 10.1016/j.rmed.2021.106555; 10.3390/DIAGNOSTICS11091647; 10.1007/s10877-021-00651-x.
Overall, I think this work is excellent.
Author Response
Point 1: In table 1, it would be useful to add the patients comorbidities in the “characteristic” column, if these data are available.
Response 1: We have added the patients' comorbidities to Table 1.
Point 2: The reference list should be implemented with the following papers: 10.1016/j.rmed.2021.106555; 10.3390/DIAGNOSTICS11091647; 10.1007/s10877-021-00651-x.
Response 2: We have updated the reference list as suggested.
Reviewer 2 Report
The authors have investigated the effect of individualized PEEP determined using electrical impedance tomography in post-cardiac surgery patients. The paper can be accepted provided some revisions are carried out as follows:
- In introduction, kindly provide disadvantages of existing methods.
- In methodology clearly state the inclusion criteria. how was ovariectomy confirmed? Page 8 line 50-51 shows “12 weeks after surgery, application of femur was for verification of ovariectomy.” This sentence needs reframing and better clarification. When was treatment started? For how long?
- In discussion, kindly add few advantages of EIT-guided individualized PEEP. Also a section discussing whether individualized PEEP can be extended to other kind of surgeries excluding cardiac surgery would increase readability and will be interesting.
Author Response
Point 1: In introduction, kindly provide disadvantages of existing methods.
Response 1: As you suggested, we have added the disadvantages of existing methods to the introduction section. (page 1-2, lines 43-46)
Point 2: In methodology clearly state the inclusion criteria. how was ovariectomy confirmed? Page 8 line 50-51 shows “12 weeks after surgery, application of femur was for verification of ovariectomy.” This sentence needs reframing and better clarification. When was treatment started? For how long?
Response 2: This sentence does not appear to be in our manuscript.
Point 3: In discussion, kindly add few advantages of EIT-guided individualized PEEP. Also a section discussing whether individualized PEEP can be extended to other kind of surgeries excluding cardiac surgery would increase readability and will be interesting.
Response 3: We have made some revisions to the discussion, adding advantages of EIT-guided individualized PEEP and suggesting that individualized PEEP for other types of surgeries can be beneficial (page 9, lines 333-335).